# A Simulation-Based Clinical Nursing Education Framework for a Low-Resource Setting: A Multimethod Study

**DOI:** 10.3390/healthcare10091639

**Published:** 2022-08-28

**Authors:** David Abdulai Salifu, Yolande Heymans, Christmal Dela Christmals

**Affiliations:** 1Centre for Health Professions Education, Faculty of Health Sciences, North-West University, Potchefstroo Campus, Building PC-G16, Office 101, 11 Hoffman Saint, Potchefstroom 2520, South Africa; 2School of Nursing and Midwifery, University for Development Studies, Box TL 1350, Tamale 00233, Ghana

**Keywords:** clinical competence, clinical nursing education, low-resource setting, sequential multimethod, simulation, simulation-based clinical nursing education

## Abstract

Simulation-based clinical education is a useful strategy for teaching, learning, and assessing clinical competence in health professions education. However, the use of simulation-based clinical nursing education (SBCNE) in low-resource settings such as Ghana has been hampered by the lack of a context-specific framework to guide its design, implementation, and evaluation. This study sought to develop a context-specific framework to guide the design, implementation, and evaluation of SBCNE in a low-resource setting. The study employed a sequential multimethod design, comprising a scoping review; qualitative descriptive design (situational analysis) made up of two parts–focus group discussions (FGDs) with post-registration nurses and nursing students, and semi-structured interviews with nurse educators; and narrative synthesis of the scoping review and situational analysis data, used to develop a draft SBCNE framework for a low-resource setting. The draft SBCNE framework was evaluated by stakeholders of nursing education and practice using nominal group discussions. The framework is comprised of five constructs (context, planning, design, community of learning, and outcomes). The user-centric, comprehensive, context-specific SBCNE framework has the potential to enhance the implementation of simulation in nursing education and the development of clinical competence in a low-resource setting. As a result, we urge nursing leaders and nurse educator unions to take the lead in lobbying regulatory bodies, the central government, and their development partners to provide the necessary financial support and resources for the implementation of the framework and adoption of SBCNE in low-resource settings.

## 1. Introduction

Globally, nursing education institutions (NEIs) are progressively adopting competency-based curricula and the use of simulation-based clinical nursing education (SBCNE) as strategies to facilitate competence development [1,2,3,4,5]. Many stakeholders of nursing education, particularly in developed countries, have advocated and, sometimes, developed frameworks or theories to guide the design, implementation, and evaluation of SBCNE [6,7,8]. Historically, the military and aviation industries have long used simulation to train pilots and other technical staff in both skills and safety procedures [9]. This is because decreasing clinical placement facilities, increasing student enrolment numbers, poor clinical supervision, lack of resources, and ethical concerns about using patients for practice have all impeded the development of clinical competence in the clinical practice setting (CPS) [10,11,12]. Furthermore, the current clinical skills teaching strategies (demonstration and lectures) commonly used by nurse educators, especially in low-resource settings, are ineffective in fostering clinical competence development [11,12,13,14]. Competency-based curricula and SBCNE are intricately linked [15]. Therefore, SBCNE is essential in filling these gaps and in providing nursing students with the opportunity to develop clinical competence.

Clinical simulation seeks to replicate real clinical situations in the simulation laboratories of NEIs in a safe and non-threatening learning environment for students [8,16]. Many people think of simulation as the use of high-fidelity computerized manikins that mimic normal physiological function [9]. However, it is important to keep in mind that using low-fidelity simulations (LFS), such as role-play, has similar benefits to those of high-fidelity simulations (HFS) [9,17]. To ensure no setting is left behind in the use of SBCNE, the WHO has suggested the use of high-fidelity simulation (HFS) for resource-endowed settings such as first-world countries and low-fidelity simulation (LFS) for low-resource settings [18,19]).

SBCNE is considered a valuable approach for teaching, learning, and evaluating clinical competence in health professionals [8,20]. If effectively combined with clinical placement experiences and immersive student-centred teaching and learning strategies, SBCNE can be a powerful approach to facilitating the development of clinical competence [21,22,23,24,25]. Empirical evidence lends credence to the fact that SBCNE is beneficial for students, patients, and the overall health system. The positive impact of SBCNE is evident in its ability to enhance students’ confidence, critical thinking and clinical reasoning skills, and clinical competence, ensuring that competent nurses are produced for quality health care and positive patient outcomes [21,22,23,24,25]. Hence, the use of SBCNE has been strongly recommended by the National Council of State Boards of Nursing (NCSBN) and the World Health Organization (WHO). Nonetheless, the use of SBCNE in low-resource settings has been slow to progress, partly due to the lack of a context-specific framework to guide the design, implementation, and evaluation of simulations, as well as deliberate investment to transform and scale up nursing education [26,27].

Prior research has emphasized the need for an empirical framework to guide the design, implementation, and evaluation of simulation in nursing and medical education [27,28,29]. It is no doubt that, a comprehensive simulation framework with well-defined constructs ensures a well organised and purposeful simulation experience [8]. A few frameworks [8,30,31,32,33] have been developed to guide the design, implementation, and evaluation of SBCNE. These frameworks were useful in resource-endowed settings as they were developed and tested within first-world countries. The current study therefore aimed to address this gap by developing a context-specific SBCNE framework to guide the design, implementation, and evaluation of simulation in NEIs in low-resource settings. In addition to developing the context-specific SBCNE framework for low-resource settings, the study also reiterates the need to strategically invest, transform, and scale up nursing education in low-resource settings.

## 2. Materials and Methods

### 2.1. Design 

This study employed a sequential multimethod research design conducted in four phases [34,35,36]. In this study, the use of a sequential multimethod design enabled the combination of four research designs (a scoping review, qualitative descriptive study, narrative synthesis, and nominal group discussions) that are complete studies on their own in one large study [37]. In this study, phase 1 (scoping review), phase 2 (situational analysis–qualitative descriptive study), phase 3 (development of the framework–narrative synthesis) and phase 4 (evaluation of the framework–nominal group discussions) are complete studies on their own. The phases of the study were guided by the six milestones of research and design methods [38,39]. The six milestones include: to identify the problem motivating the framework development; describe the objectives of the framework; design and develop the framework; subject the framework to testing; evaluate the results of testing; and communicate the results (finalized framework). 

#### 2.1.1. Phase 1: Scoping Review

Phase one had two objectives: 1. To identify the constructs of frameworks and theories used to guide the design, implementation, and evaluation of simulation in nursing education globally; 2. To describe the applicability of the constructs of simulation frameworks and theories used to guide the design, implementation, and evaluation of simulation in nursing education globally, in the context of low-resource settings. Objective one provided a clear understanding of the various constructs of simulation frameworks and theories used to guide the design, implementation, and evaluation of simulation in nursing education globally. Objective two helped in the design of the primary studies of phase two. Details of the scoping review are published in another journal [27]. 

#### 2.1.2. Phase 2: Situational Analysis–Qualitative Descriptive Study 

Phase two was conducted in two parts. The first part entailed a qualitative descriptive study comprising of FGDs of 15 post-registration diploma nurses and 40 nursing students (20 second- and 20 third-year students) from six different sites (three NEIs, which were all accredited diploma-awarding public nursing colleges and their primary clinical sites) in three geographical zones of Ghana, a low-resource setting [12]. The second part comprised of semi-structured interviews with nine nurse educators from the three NCs [40]. Purposive sampling technique was used in recruiting the participants [41] to ensure the selection of persons with rich information. See Table 1.

Data from the situational analysis were analysed using the framework approach of data analysis [42] with the aid of ATLAS.ti. The findings of the situational analysis are published in two different papers [12,40].

#### 2.1.3. Phases 3 and 4: The Development and Evaluation of the Framework

The results of phases 1 and 2 were narratively synthesized, satisfying the first and second milestones of the six milestones of research and design methods [38,39] and served as the data for phase 3 (development of the draft framework), which achieved the third milestone. The National League for Nursing (NLN) Jeffries Simulation Theory [8] was adopted as the conceptual framework for organizing the constructs of the SBCNE framework for low-resource settings. The NLN Jeffries Simulation Theory is anchored on the constructivist and experiential learning theories. According to the constructivist learning theory, learning is constructed through the active development of unique perspectives and knowledge of the world through experience and reflection [43]. Experiential learning theory, on the other hand, holds that knowledge is created through an iterative process involving the transformation of experience in four adaptive learning modes: concrete experience (feeling), reflective observation (observing), abstract conceptualization (thinking), and active experimentation (doing) [44]. Components of the theory include context, background, design, simulation experience, facilitator and educational strategies, participant, and outcomes. The context, background, and design components of the theory could be classified as the planning phase of the simulation activity. Jeffries [8] defined context as the actual location of the simulation activity, whether it takes place in a school or a clinical setting, as well as if the simulation is for educational or evaluative purposes. Background includes the proper integration of simulation activities into the curriculum as well as the identification and allocation of necessary resources for the successful implementation of the simulation. The simulation design characteristics includes; the development of specific learning objectives, the simulation scenario with varying degrees of complexity and fidelity, roles, and debriefing [8]. The planning phase precedes the simulation experience which is marked by a dynamic interplay between the facilitator and participant, as well as the application of effective educational strategies. The simulation outcome is multifaceted with benefits targeted at participants, patients, and systems [8]. 

The findings of phases 1 and 2 were narratively synthesized to produce consolidated evidence for the development of the framework. Narrative synthesis entails the use of textual descriptions to generate consolidated data from different qualitive enquiries [45]. Findings of the scoping review (phase 1) and situational analysis (phase 2) were coded iteratively according to the constructs of the National League of Nurses’ (NLN) Jeffries [8] Simulation Theory. The codes were compared to reach consensus and refined. Where the data did not appropriately fit into any of the identified constructs, new constructs were created to accommodate the emerging complexity [45] (Appendix A—provided as a Appendix A). Findings of the narrative synthesis were then used to develop and diagrammatize a draft context-specific SBCNE framework to guide the design, implementation, and evaluation of simulation in NCs in Ghana, a low-resource setting. 

The draft framework from phase 3 served as the data for phase 4 (evaluation of the draft framework). The evaluation and finalization of the framework were achieved in phase 4 of this study through nominal group discussions. The draft framework was evaluated for its appropriateness in facilitating the development of clinical competence by nursing students, nurse educators and post-registration nurses. Nurse educators and nursing students of accredited public nursing colleges, in Ghana, and post-registration nurses working in the primary clinical sites of the chosen NCs who participated in phase 2 (situational analysis) of the study were eligible to participate in this phase of the study. A purposive sampling technique was used to recruit a minimum of six (6) nurse educators (2 per college) with a minimum of three years’ experience in practical skills teaching; eight (8) nursing students (at least, 2 per college) with a minimum of one year experience as a student; and a minimum of six (6) post-registration nurses (2 per hospital) within their first year of practice and working in a general ward for this phase of the study. These categories of participants were selected because they were part of the problem identification phase (situational analysis) of the study. Hence, some of the solution they proffered in the situational analysis were factored into the draft framework. Therefore, their role in the evaluation of the draft framework cannot be overemphasized. Nurse educators, nursing students, and post-registration nurses who chose to withdraw from phase 2 of the study or failed to sign the informed consent form were excluded from this phase of the study. Varying ways of evaluating educational frameworks and models have been reported in the literature. Christmals and Armstrong [34] evaluated their curriculum framework developed for sub-Saharan Africa using experts from various universities in sub-Saharan Africa. In Dobbie et al. [46], Lloyd-Jones et al. [47] and Anim-Boamah [48], Nominal Group Discussions (NGDs) were used to evaluate the curriculum, medical undergraduate program and a clinical competency assessment framework, respectively. An NGD involves the use of a more structured format of brainstorming to obtain divergent information from a group of people on a phenomenon [49]. The use of the NGD ensured the content of the framework was interrogated from different points of view, as well as enabling the prioritization of ideas [50]. Open discussions were encouraged and the ideas that emerged were used to revise the framework. The adoption of the NGD technique was fit for purpose given the need to ensure the SBCNE framework was low-resource-context specific. To ensure homogeneity and foster active participant engagement, the nominal group discussions were held separately for each of the groups using a topic guide (Table 2). Suggestions from the stakeholders were used to finalize the framework. Additionally, through informal contacts, the SBCNE framework was submitted to four nurse educators with duo experience in clinical nursing education and simulation in both developed and low-resource settings to assess for face validity. Inputs from these experts were considered in the final design of the framework.

Eligible participants were nurse educators, students, and post-registration nurses who participated in the semi-structured interviews or FGDs in phase 2 of the study who were purposefully selected. Due to COVID-19, the study was conducted via Zoom. The data (field notes and audio recordings) collected from the three groups (nurse educators, nursing students, and post-registration nurses) were analysed using the framework approach of thematic analysis [42] with the aid of ATLAS.ti software to compare and contrast results by categories of individual nominal groups in search for patterns and themes. Figure 1 below illustrates the research design.

### 2.2. Ethics Approval 

The ethics approval for this study was granted by the North-West University Human Research Ethics Committee (NWU-00431-20-A1) and the Ghana Health Service Ethics Review Committee (GHS-ERC019/08/20). The researchers observed ethical requirements during the different phases of the study. All participants gave their informed consent and duly signed informed consent forms during the different phases of the study before the start of data collection. To preserve participant privacy and anonymity, codes were assigned to the participants prior to data collection and were utilized throughout the data collection and analysis process. Participants were informed that participation in the study was entirely voluntary, and that they could withdraw at any time without penalty.

### 2.3. Rigor

To ensure the credibility of the scoping review, the first and third authors D.A.S and C.D.C independently evaluated the retrieved articles by reading the titles and the abstracts; the second author, Y.H served as an adjudicator in instances where D.A.S and C.D.C failed to reach a consensus. All data extraction sheets and interview guides were submitted to experts in qualitative research and scholars in teaching and learning in nursing education for review. The inputs from the experts were factored into the redesign of the instruments. Moreover, the instruments were pretested in an analogous institution with post-registration nurses, nurse educators, and nursing students before use. With reference to data analysis, co-coding was carried out. During the data analysis process D.A.S and C.D.C coded two scripts of the qualitative data independently, compared the codes to reach consensus and refined the codes which were then applied by D.A.S in coding all the transcripts. This was carried out for all the data sets in phase 2 (focus group discussions and individual interviews) and phase 4 (nominal group discussions) of the study. Moreover, the transcripts, codes, and thematic framework of all the qualitative data were verified by D.A.S and C.D.C for their appropriateness. D.A.S. and C.D.C independently synthesized the findings of the scoping review and situational analysis. Dialogue was used to settle disagreement between the two researchers. Where it was difficult to reach consensus, the third researcher served as an arbitrator. The researchers had no personal relationship with the study participants. However, independent research administrators were used in recruiting participants for the FGDs, individual interviews and nominal group discussions. To ensures findings relating to the qualitative data of the study were free from the personal biases and prejudice of the researchers, the concept of bracketing was applied. The researchers and independent focus group facilitator proclaimed personal biases and assumptions of the phenomenon under study and set them aside [51,52]. To enhance rigour, we used the concept of member checking [42] to confirm the modifications recommended during the nominal group discussion. The findings of the nominal group discussion were presented to selected participants through a Zoom session to confirm the accuracy of the results. No discrepancies arose during the discussion. No repeat interviews or discussions were held for findings of phase 2. 

## 3. Results

### 3.1. Phase 1: Scoping Review

Seven constructs were identified and described: context, background, simulation design, educational practices, facilitator, participant, and outcome. The review revealed a lack of a context-specific framework to guide the design, implementation, and evaluation of SBCNE in low-resource settings. Moreover, there were gaps in applying simulation-based framework(s) developed in first-world countries to low-resource settings. Peculiar challenges that appear to confront the implementation of SBCBE in low-resource settings such as resource limitations, the lack of well-trained simulation facilitators, and large student numbers are not addressed in the existing simulation frameworks and theory. The findings of the review therefore underscored the need for the development of a context-specific framework tailored to the needs and resources of low-resource settings, to promote the use of SBCNE. The scoping review is published as a separate paper [27].

### 3.2. Phase 2: Situational Analysis–Qualitative Descriptive Study 

Three themes emerged from the FGDs: nursing education institutional factors; clinical placement design, implementation, and system challenges; and challenges of clinical teaching and learning, whereas four themes, namely, nurse educator and student factors; skills learning environment factors; institutional challenges; and regulatory issues emerged from the semi-structured individual interviews. We found that the current approach to clinical nursing education, such as the over-reliance on clinical placement and the use of more teacher-centred teaching approaches, were ineffective in the teaching and learning of practical skills and the development of clinical competence [12,40]. Moreover, large student numbers, the lack of major educational resources and incentives for nurse educators’ career enhancement and professional development were found to hinder the teaching of clinical skills and clinical competence development in Ghana [12,40].

### 3.3. Evaluation of the Framework: Nominal Group Discussion

A total of 20 participants comprising eight second- and third-year nursing students, six post-registration nurses, and six nurse educators participated in NGDs to evaluate the draft SBCNE framework for low-resource settings. The draft SBCNE framework was evaluated under three main categories. Category 1 addressed appropriateness, acceptability, and applicability, Category 2 suggested modifications, and Category 3 described the development process. The results of the NGDs yielded 19 codes which were grouped into six sub-categories and then regrouped into the three main categories: appropriateness, acceptability, and applicability; suggested modifications, and development process. All the participants of this phase of the study were part of the situational analysis (phase 2) of the broader study. The demographic information of the participants is contained in Table 3. 

#### 3.3.1. Category 1: Appropriateness, Acceptability, and Applicability

According to the participants, the framework was properly designed and structured to match the needs and resources of a low-resource setting. All components of the framework were thought to be well-connected and important in ensuring the successful design and implementation of SBCNE in a low-resource setting. None of the components of the framework, according to the participants, should be deleted. As a result, they thought the framework was adequate for guiding the design, implementation, and evaluation of SBCNE for clinical competence development in a low-resource setting. For example, one post-registration nurse had this to say about the appropriateness of the draft SBCNE framework for low-resource settings:
*“…I strongly believe the framework is well structured, all the components of the framework are well related”*. NGDPRN1
*“I think if the framework is well implemented it is going to produce an excellent result at the end of the day, its objective of equipping nursing students with clinical competence is going to be achieved”*. NGDPRN3

Because of the self-explanatory and easy to interpret and relay nature of the framework, participants anticipated it would be well received in the setting. They also thought the structure was appropriate for the situation. Some participants, on the other hand, suggested that in order for the framework to be properly implemented, regulatory bodies must commit to using it. 

#### 3.3.2. Category 2: Suggested Modifications

Although the participants agreed all the components of the framework were important and that none should be deleted, some suggested adjustments to the structure and nomenclature of some of the components of the framework to ensure specificity. In addition, one participant with prior experience in simulation proposed that the simulation community of learning be expanded to include other members, such as other nurse educators without simulation facilitator roles, other participants (other students either than the class to experience the clinical simulation), clinicians, and regulatory bodies with influence in the participant learning. The participants also proposed adjustments to the graphical illustration (structure) of the framework to improve its suitability and genuine reflection between the content and structure.


*“The teaching and learning strategies aren’t living entities to co-exist with facilitators and participants in a team bound by mutual trust”. This relationship could be illustrated by expanding the community of learning to include the other members such as clinicians, other students, and other nurse educators as the three actors bound by the mutual respect and teamwork”.*
NGDNE1


*“If learning has to be sustainable, the community of learning has to include other students, clinicians, the regulatory bodies, and all other persons who the students interact with during their learning journey”.*
NGDNE1


*“Teaching and learning strategies should also be renamed as immersive teaching and learning strategies for the purpose of specificity and placed in the middle of the community of learning together with teamwork”.*
NGDNE1

The inclusion of clinicians in the simulation community of learning, according to the participant, has a larger potential for effectively educating student nurses to face the reality of clinical practice. These inputs coming from the participant could be as a result of his prior exposure to the use of simulation in nursing education. 

#### 3.3.3. Category 3: Development Process

Participants were in agreement that the approach adopted in the development of the framework—conducting a scoping review to identify and describe constructs of existing frameworks and theories used in nursing education around the world, focusing on their applicability in a low-resource context, as well as a situational analysis to explore the experiences and perceptions of nursing students, post-registration nurses, and nurse educators was collaborative, consultative, and appropriate. 


*“Well, with how it was developed that was how actually it should be done. You involved people who are into it, the framework is designed to develop the students, and in developing these students, it is the nurse educators that are supposed to help the students develop those competencies, and you spoke to both students and the nurse educators and went ahead and reviewed the literature that was available. So, the development process of the framework was okay and I think is good”.*
NGDPRN4


*“Since it’s a framework you are trying to develop, I think doing a scoping review to find out what already exists and what has already been done in the field and also requesting the views of the implementers, that is the nurse educators and the students, would be a good idea. It is good because you will be able to draw a comparison”.*
NGDNE2

Participants saw a user-centric approach as the best and most appropriate way to develop the framework.

### 3.4. The Simulation-Based Clinical Nursing Education Framework for a Low-Resource Setting 

Section 3.4 presents the final proposed framework informed by findings of the narrative synthesis of the scoping review and situational analysis. The final SBCNE framework consists of four broad components: planning, design, community of learning, and outcomes, which exist within the context. The bi-directional arrows linking the four main components of the framework denotes the reciprocal influence of the components on each other. The outcomes are greatly influenced by the degree of effective attainment of each of the other broad components. Thus, the level of attainment of the outcomes could possibly suggest a manipulation of the other components to maximise the benefits of the simulation experience. The components of the framework, as illustrated in Figure 2, are described below.

#### 3.4.1. Context

The “context” is an important component of the SBCNE framework. It refers to the broader environment of the simulation activity, whether it occurs in a high- or low-resource setting, and more precisely, whether it occurs in a school or clinical setting [8]. The availability of logistics and human resources to support the simulation activity are two crucial contextual aspects to consider. The general objective of the simulation activity, whether it is for instructional or evaluative purposes, is also included in the context.

#### 3.4.2. Planning

The adoption of SBCNE as a pedagogical strategy requires thorough planning. Given the novelty of the concept of SBCNE in low-resource settings, careful planning is required to ensure the successful design, implementation, and evaluation of the concept. A detailed needs assessment of the availability of the appropriate resources is required as part of the planning process to ensure the success of the simulation experience.

##### Needs Assessment

Given the variation in the available resources in different settings and NEIs, a needs assessment is required to determine the actual level of resources available to support the simulation activity. As part of the needs assessment, the physical infrastructure and the human and material resources of the educational institution wishing to implement the simulation programme are all assessed in depth [8,30,53]. Inadequacies discovered during the needs assessment are then rectified to ensure a successful simulation experience.

##### Skills Laboratory Retooling

This framework recommends the need for transforming and scaling up the skills laboratories of low-resource settings to at least be able to support low-fidelity simulation. Expansion of skills laboratory space and the provision of essential consumables and equipment to support real clinical experiences are encouraged.

##### Preparation/Training/Orientation

Adequate preparation, training, and orientation of simulation facilitators (nurse educators), participants, and other members of the simulation community of learning are essential pre-requisite for the successful implementation of the simulation.

##### Facilitators

Central to the successful implementation of simulation is the need for adequate training of nurse educators to act as simulation facilitators with the capacity to design scenarios, set up manikins, and facilitate the simulation sessions. However, the major hurdle to the successful implementation of simulation in nursing education, particularly in low-resource settings, has been identified as insufficient staff training [54]. In low-resource settings, without the adequate training of nurse educators—even with the purchase of high-fidelity manikins—they risk only serving the purpose of low-fidelity manikins. Adequate training of nurse educators is therefore crucial to the successful implementation of SBCNE in low-resource settings.

##### Participants and Other Members of the Simulation Community of Learning

An orientation programme in the form of seminars or workshops is required to create awareness of the simulation activity among simulation participants and other relevant members of the simulation community of learning, such as clinicians and other nurse educators. Such orientation events are designed to reduce anxiety, especially among students, and promote acceptability, sustainability, and continued learning.

##### Curriculum Integration

Adequate curriculum integration is an essential requirement for the successful implementation of simulation. The simulation experience must be well-structured and matched with the course material and learning objectives as specified in the programme curriculum for effective design and implementation [8,30]. Arthur et al. [16] recommends the integration of simulation into all clinical courses in the curriculum with a progressive level of complexity.

#### 3.4.3. Design

The simulation design is an element that exists within context and has a significant impact on the simulation activity. The contextual elements which are identified through the needs assessment are often used to direct the formulation of the broad simulation goals, which in turn influence the design of the learning objectives [8,30,53].

##### Learning Objectives

An essential prerequisite for the successful design and implementation of the clinical simulation experience is the establishment of appropriate, clear, concise, and measurable learning objectives that are linked to the curriculum and in line with the participants’ educational level and needs [8,30]. The objectives should be reasonable in number, at least three to four, and be focused on the three domains of learning (cognitive, psychomotor, and behaviour) [55]. The learning objectives are expected to direct the simulation design characteristics such as scenario development, including the level of complexity and fidelity, the choice of simulation modality, and the cues to support participants during the simulation experience. The simulation learning objectives should be formulated and pre-discussed with the participants in the pre-briefing session before the commencement of the clinical simulation experience. This will help in keeping the participants focused on achieving the broader objectives of the clinical simulation experience.

##### Scenario Development

The need to develop a scenario in line with the learning objectives and overall outcomes of the simulation is an essential prerequisite of the simulation design. The simulation scenario should detail a case report indicating the patient’s condition, health problems, prescribed medications, and other important information. If role play is going to be used, then the actor script also needs to be developed as part of the scenario. The simulation scenario establishes the appropriate context for the simulation experience. The simulation scenario should consist of varying degrees of complexity and fidelity to foster critical thinking [8]. Fidelity refers to the degree to which the simulation experience mimics reality, and it is influenced by the environment, equipment, and student factors [8,16,19]. To achieve conceptual fidelity, all the components of the simulation scenario should relate to each other in a realistic manner that allows for the participants to construct meaning; for example, the patient complaints must relate to the disease condition [8,30,31]. Moreover, the simulation scenario should have the ability to suspend disbelief in participants in meeting the demands of psychological fidelity. Physical or environmental fidelity refers to how the immediate environment of the simulation experience replicates the real clinical environment of the setting, where the use of artefacts is encouraged to enhance moulage [16,53]. Cues are established and designed as part of the scenario development process to provide clues to participants as they progress through the simulation clinical experience [8].

##### Simulation Modality

The clinical simulation experience needs to be structured in line with the simulation modality [53]. The simulation modality is described as the simulation method used to implement the simulation scenario. Due to resource limitations, this framework recommends the use of low-fidelity methods such as manikins, role play, or task trainers based on the resource strength of the setting for the implementation of simulation in low-resource settings. Students build competence by acting out simulated scenarios in role play. For invasive procedures such as intramuscular and intravenous (IV) injections, task trainers (static manikins) such as IV arm and injection pads should be used if available.

##### Pre-Briefing

The need for pre-briefing and a more thorough orientation for participants prior to the simulation experience is crucial for the successful implementation of the simulation experience. The pre-briefing should be a well-structured information session held before the start of the simulation experience to familiarize students with the learning objectives, ground rules, role assignments, space, equipment, and simulation modality to be used [8,30,33]. This frequently reduces participant nervousness and produces a calm environment in which the simulation experience can begin easily. The orientation period also offers an opportunity for dress rehearsals for selected participants engaged in role play or other important aspects of the simulation experience before the commencement of the simulation sessions [30].

##### Videography

This framework proposes the inclusion of videography as a component of the simulation design. For videography to be successfully incorporated into the clinical simulation experience, proper sensitisation of participants on the use of videography and adequate training of a selected person to act as a videographer are essential. However, more research is needed to guarantee the successful use of videography in simulation design and debriefing especially in low-resource settings.

##### Structured Debriefing

Debriefing is an indispensable component of the simulation activity, with a direct influence on the clinical simulation experience. To be effective, the debriefing session should be conducted soon after the simulation experience and should last as long as the clinical simulation experience. The debriefing session should be well-structured, learner-centred and non-threatening to the participants [16]. Debriefing enables participants and the facilitator to reflect on their actions during the simulation activity. The debriefing session should be more focused on reviewing the clinical simulation experience with the aim on clarifying and building on the knowledge and skills gained from the clinical simulation experience. This framework recommends the use of video recordings to guide the debriefing session. Video-enhanced debriefing will offer participants the opportunity to review, analyse, and reflect on their actions, resulting in self-awareness and reflective learning, thereby promoting critical thinking and the mastery of clinical skills. For debriefing to be effective, an evaluation needs to be performed by comparing the actual performance of the participants with the desired learning objectives that were set at the beginning of the simulation experience [16]. It is essential for the debriefing session to be well structured to inspire reflective thinking and identify areas of weakness for deliberate practice and peer learning [56]. There are various techniques to structured debriefing, but this framework suggests Coutinho et al.’s [57] four-phase approach, which includes meeting, positive reinforcement, analysis, and synthesis.

#### 3.4.4. Community of Learning

The need for the establishment of an effective simulation community of learning is pivotal to the success of the clinical simulation experience. The simulation community of learning is characterised by a dynamic interplay between facilitators who are expected to be knowledgeable in simulation and debriefing, participants with the right attitude towards learning, as well as other members of the broader simulation community of learning (clinicians, other nurse educators, other participants, and the regulatory bodies) bound by the use of immersive teaching and learning strategies and existing within an environment of effective teamwork [8,58].

##### Facilitator

The facilitator is described as a nurse educator with the knowledge and abilities to provide support to participants during the simulation activity. The nurse educator’s expertise in simulation is required for the successful design and implementation of simulation. The nurse educator must be knowledgeable in simulation, pre-briefing, and debriefing [8,31,33]. Rather than being a “sage on the stage”, the facilitator must act as a “guide on the side”, adopting more learner-centred strategies and creating a non-threatening atmosphere for participants. The facilitator offers guidance and support for participants throughout the simulation experience. The facilitator’s background, including as years of experience, clinical expertise and a positive attitude, are believed to be closely associated with the success or otherwise of the simulation activity [8].

##### Participant

The participant is a person who comes into the community of learning with the objective of gaining knowledge and skills. The participant is expected to be a learner who is self-directed and motivated (Billings & Halstead [59]) and to set their own objectives or expectations and seek to achieve them with the help of the facilitator. The setting of high expectations enhances the clinical simulation experience and ensures the attainment of the simulation goals [8]. Depending on the simulation modality to be used, the participant could assume different roles during the clinical simulation experience [31]. Such roles could include taking part in a role play where the participant could be required to act as a patient, a nurse or a patient relative, a video recorder, or an observer. Facilitators must ensure role rotation to ensure that all participants benefit from the simulation experience. The roles should be properly discussed and assigned during the pre-briefing session, and the participants should be expected to rotate roles during the simulation experience. Other participant variables with direct influence on the success of the simulation experience include the participants’ program level and attitude towards learning [8]. The program level corresponds with the content area and determines or directs the level of complexity of the simulation experience. Moreover, the success of the simulation experience appears contingent on the good attitude of the participant towards learning.

##### Other Members

Including other members of the broader learning community of nursing education (other participants, clinicians, and nurse educators, and regulatory bodies such as the Nursing and Midwifery Council (NMC) and the Ministry of Health (MOH)) in the simulation community of learning will help enhance the positive impact of SBCNE and its sustainability in low-resource settings. This could be achieved through effective collaboration between the management of the NEIs and that of the clinical settings, together with the MOH and NMC leading the policy direction. The inclusion of clinicians in the simulation community of learning will guarantee uniformity in the performance of nursing procedures in both the school and clinical learning environments, thereby addressing the issue of the theory–practice gap. This is expected to reduce the stress on simulation facilitators and participants and ensure sustained learning.

##### Immersive Teaching and Learning Strategies

This simulation framework adopts more immersive learner-centred and experiential teaching and learning strategies. Literature is rife with the negative impact of teacher-centred teaching and learning strategies.

##### Interactive Learning

The learner readily loses concentration when the teaching and learning activities fail to be engaging and interactive for the learner [59]. Adopting more learner-centred, immersive, interactive, experiential, interprofessional, and cooperative strategies in establishing a more conducive learning environment has proven to be more effective in promoting the critical thinking skills of learners [8,59]. To actively engage the participant in the construction of their own knowledge during clinical simulation experience, this framework adopts an interactive and experiential learning strategy. The key element of experiential learning is its ability to actively engage participants in the construction of their own knowledge by enabling them to experience things and reflect on them [60]. Moreover, the use of interactive teaching and learning strategies such as role-play enables the facilitator to adequately evaluate the critical thinking and problem-solving skills of participants during the simulation activity [8].

##### Station Teaching

It is instructive to note that the appropriate number of participants to be engaged in a simulation experience is largely determined by the simulation objectives. That notwithstanding, for the simulation activity to be meaningful and impactful, the participant number in a simulation session should be small enough to ensure every participant has a good chance to be actively involved in the simulation experience [61]. In a study exploring students’ perceptions of their learning experience using high-fidelity simulation, Partin et al. [62] found that the students were unhappy when participants in a simulation session numbered more than six. This finding may be construed to suggest that the number of participants in a simulation session should not exceed six. However, given the large number of students enrolled into NEIs in low-resource settings, restricting the number of participants in a simulation session to six runs the danger of preventing a substantial number of the students from participating in the simulation experience.

##### Time on Task

Time on task is part of the process of immersive teaching and learning strategies. The appropriate use of time by both the facilitator and participants is important for the success of the simulation experience. Enough time should be assigned for the simulation experience and factored into the course timetable. A number of studies have confirmed the correlation between longer hours of simulation sessions and the improvement of learning outcomes [63,64]. Unfortunately, none of the studies indicated the ideal duration for a simulation session. This framework recommends that a whole day be set aside in the timetable for simulation. The time used for each simulation session could be maximised by assigning realistic time frames to tasks during the simulation. Familiarizing participants with the simulation ground rules, the equipment, physical environment, and learning objectives during the pre-briefing/orientation session could help to keep the simulation experience focused, thereby promoting efficient use of time. Considering the importance of pre-briefing, the simulation activity, and debriefing, Jeffries [65] recommends that each of the three areas should be given equal attention. More research is required to establish the right timing for a simulation session based on the objectives, scenario, and purpose of the simulation.

##### Guided Reflection

Reflection should not simply be a recall of events, but rather a detailed examination of the simulation experience to identify and correct any flaws or mistakes that may have occurred during the simulation. Given that the simulation participants in low-resource settings are novices, the use of guided reflection will provide a better opportunity for effective reflection. It will be difficult for inexperienced participants to reflect freely without support. Guided reflection is described as a well-structured process of reflection that occurs between the facilitator and participant in a simulation experience [66].

##### Deliberate Practice

The use of deliberate practice in this framework refers to the act of engaging in self-learning and repetitive practice in the simulation laboratory with the aim to improve psychomotor skills and clinical competence [56,67]. Following debriefing, feedback, and reflection, deliberate practice is required to ensure an effective simulation experience, allowing for the mastery of psychomotor skills. Deliberate practice strives to strengthen areas of strength while also addressing areas of weakness. The utilisation of deliberate practice in simulation laboratories is said to be effective in supporting students to develop clinical competence [67].

##### Peer Learning

Peer learning is a learning strategy incorporated in this framework to ensure the adoption of varied strategies by participants to facilitate their own learning process. It is a contemporary learning approach in education in which learners take control of their own learning by supporting each other through the learning process [68]. The prerequisite for peer learning in this framework is debriefing, feedback, and reflective thinking. These three items enable the identification of challenging procedures following the simulation experience for peer learning. The adoption of peer learning in SBCNE offers the opportunity for participants to share their unique experiences and guide each other through their areas of weakness [68]. There is enormous research evidence to support the use of peer learning in SBCNE [69].

##### Teamwork

Teamwork is the binding force with a direct influence on the interplay between the facilitator, participant, and other members of the simulation community of learning. For the successful implementation of SBCNE, there must be effective teamwork among the key actors in the simulation community of learning. The components of teamwork include mutual trust and collaboration, support, and feedback. The relationship between these key actors holds the potential to influence the simulation experience, attainment of the simulation learning objectives, and the sustainability of the SBCNE. Therefore, there must be effective mutual trust and collaboration between members of the simulation community of learning. The establishment of mutual trust enhances effective collaboration and promotes the free flow of information. Such a collegial environment enables participants to freely ask questions and actively participate in the learning process to acquire knowledge and skills.

##### Support

Adequate support is required to ensure a more impactful simulation experience for participants. The cues created during the scenario development are used as prompts to support participants during the simulation experience. They should provide enough detail to help participants through the simulation experience without interfering with their independence or capacity to solve problems. Participants acting as patients or patient relatives could be used to act out the cues to draw the attention of the one acting as a nurse to an ignored pressing need of the patient.

##### Feedback

Feedback serves as a form of evaluation of the clinical simulation experience that focuses on the simulation learning objectives. Prompt feedback on the performance of the participants after the simulation experience is fundamental in promoting learning. It has been established that well-delivered feedback contributes to the boosting of participant confidence, and self-fulfilment [70,71]. The feedback feature in this framework is expected to be delivered after the debriefing session. It needs to be delivered in a more supportive and non-threatening manner rather than as a criticism. A two-way evaluation or feedback from the facilitator to the participants and vice versa ensures all challenges of the simulation experience related to both facilitator and participants are addressed ensuring a more interactive and engaging learning process [8].

##### Outcomes

The clinical simulation experience has a multifaceted impact that focuses on the participant, the patient, and the health system. Outcomes serve as benchmarks for the long-term evaluation of the entire simulation-based clinical nursing education. The participant outcome refers to the direct impact of the clinical simulation experience on the participant’s level of satisfaction, confidence, critical thinking skills, and clinical competence [9,24,72,73]. The benefit of the clinical simulation experience in relation to the patient is focused on positive self-reported patient outcomes such as satisfaction, as well as positive clinical outcomes. System outcome refers to how simulation-trained nurses contribute to cost savings (cost-effectiveness) and evidence-based practice change [8]. However, the benefit of simulation pedagogy in direct patient outcomes and its associated benefit to the health system and evidence-based practice are sparingly reported in the literature.

## 4. Discussion

The findings of this study revealed a lack of a context-specific framework to guide the design, implementation, and evaluation of simulation in NEIs in low-resource settings. The previous simulation frameworks and the NLN Jeffries simulation theory [8,30,31,32,33] were all developed in first-world countries and appear to largely focus on the use of high-fidelity simulation. This creates a lacuna in their application in low-resource settings, given the varied contextual nuances that characterize the teaching and learning of clinical skills and clinical competence development between the more resource-endowed first-world countries and the impoverished low-income countries. The situational analysis in this study revealed unique contextual nuances such as inadequate investment and the scaling up of NEIs in low-resource settings. Infrastructure and basic logistics for the teaching and learning of clinical skills and clinical competence development were inadequate. Moreover, opportunities for capacity building to enhance the competence of nurse educators were limited. In addition, the overreliance on the use of lectures and demonstrations was perceived as ineffective in the teaching and learning of clinical skills and clinical competence development, suggesting that the use of more student-centred approaches may be more effective.

To the best of our knowledge, this is the first study in a low-resource setting that attempts to develop a context-specific SBCNE framework tailored to the needs and resources of low-resource settings. The study focuses on addressing the gaps associated with the use of previous simulation frameworks and theories in low-resource settings. One striking feature of this SBCNE framework is its comprehensiveness and broadness in scope in the design, implementation, and evaluation of simulation. Unlike previous simulation frameworks and the NLN Jeffries simulation theory [8,30,31,32,33], which appear to narrowly focus on simulation as an activity, placing emphasis on simulation design, implementation, and evaluation features, this SBCNE framework broadly incorporates planning that includes a thorough needs assessment to establish the resource gap warranting augmentation before the introduction of simulation.

In responding to the call by the International Nursing Association of Clinical and Simulation Learning (INACSL) to structure clinical simulation experience in line with the simulation modality [53], the SBCNE framework thus includes simulation modality as a component of the simulation design. Previous simulation frameworks and the NLN Jeffries simulation theory [13,24,25,26,27] did not include simulation modality in the structure of their frameworks and theory. Given the focus of this framework in guiding the design, implementation, and evaluation of simulation in low-resource settings, explicitly including simulation modality and the choice of low-fidelity simulation through the use of role play, manikins, and task trainers to accomplish the purpose of the simulation is necessary [74]. Contextualizing the selection of simulation modality based on the available resources and capacity of the setting is particularly important because a large number of NEIs in low-resource settings are restricted from using simulation as an instructional method due to the expensive nature of high-fidelity simulation [17,75,76]. The purchase, setup, and maintenance of high-fidelity equipment are all capital intensive and beyond the capacity of most NEIs in low-resource settings [76]. However, previous research suggests no significant difference in the use of high-and low-fidelity simulations in facilitating clinical competence development [9,17,77,78]. Despite the World Health Organization’s recommendation to contextualize the selection of simulation modality based on the existing resources [18], it is imperative to still procure some basic logistics to support the use of low-fidelity simulation in low-resource settings due to the general lack of resources [11,79]. It is therefore advised that the state of Ghana take a special interest in nursing education and offer financial support to help reform and scale up NEIs so that infrastructure and fundamental prerequisites are made available to allow the use of at least low-fidelity simulation [33].

The establishment of an effective simulation community of learning comprising well-trained facilitators, participants (students), other members (clinicians, other participants, and regulatory bodies) co-existing within an environment of mutual trust and characterised by the use of immersive teaching and learning strategies are key requirements and a novelty of the framework. In support of this position, Botma et al. [58] in a conceptual framework for educational design at modular level to promote the transfer of learning, indicate that the building of a stronger community of learning is indispensable in the training of students with the requisite clinical competence. In a low-resource setting where the introduction of SBCNE may be a challenge, the inclusion of other members such as clinicians, other nurse educators, other participants/students, and regulatory bodies is crucial in ensuring successful implementation and sustainability. Despite the fact that some of the components of the simulation community of learning exist in other previous frameworks, particularly the NLN Jeffries simulation theory [8], none of the current simulation frameworks have unified them and characterized them as a simulation community of learning in the way that this framework does.

The SBCNE framework holds the position that the need to train and sensitize nurse educators and other members of the simulation community of learning is crucial to ensure the acceptability, implementation, and sustainability of simulation in low-resource settings. Simulation facilitators (nurse educators) must be capable of formulating simulation learning objectives, designing simulation scenarios, setting up manikins, using role play, and facilitating simulation and debriefing sessions. However, in broad agreement with the findings of this study, the lack of training and capacity of nurse educators to successfully organize simulations has been identified as one of the key challenges to the implementation of SBCNE in low-resource settings [54]. Without adequate training of nurse educators in simulation, the clinical simulation experience risks being incidental and disorganized, and thus incapable of attaining the simulation goals. This framework therefore advocates for the need for investment and key policy shifts in nursing education in low-resource settings to support the use of SBCNE. This may be achieved through the building of a stronger collaboration between key stakeholders of nursing education such as management of NEIs, the central government, NMC, and public universities with the MOH in leading the policy drive. The training of nurse educators in simulation could be achieved by local universities acting in partnership with their international collaborators to offer short programs in simulation with incentives for nurse educators to participate.

The participant is an integral component of the simulation community of learning. Similarly, other frameworks view the participant or student as a key member of the simulation experience [8,30,31,33]. In simulation, the participant is expected to come into the community of learning with some knowledge and past experience in the content area under study. Botma et al. [58] refer to this knowledge and past experiences as pre-existing knowledge. The authors define pre-existing knowledge as the combination of past-experiences and knowledge which may be driven from course work (theory) as taught in the classroom environment and expected to be completed by the learner prior to the simulation experience. Knowledge is said to be constructed when newly received information is adequately incorporated into existing mental schemata [58]. Unfortunately, most nurse educators and clinicians appear to be oblivious to this fact; hence, students are often introduced to new content without prior assessment and activation of the pre-existing knowledge [58]. To effectively promote the development of clinical competence through simulation, this framework holds the view of the importance of the need to critically assess the pre-existing knowledge and misconceptions of participants in the content area and modify them before the simulation experience. Thus, the framework recommends the adoption of pre-course work to aid in the activation of pre-existing knowledge before the commencement of the simulation experience. This is expected to make learning less stressful by reducing the amount of new information the learner is expected to receive at a time.

Given the challenge of large student numbers in low-resource settings, the framework recommends the use of station teaching with a maximum of 12 participants in a group in accomplishing the purpose of the simulation experience in low-resource settings. To maximise the impact of the simulation experience with such numbers, it is recommended that light-emitting diode (LED) screens should be installed to project the simulation activities for all participants in the simulation session to see and participate actively. With station teaching, the participants and content are divided into smaller groups; the participants should not exceed 12 in a group, and facilitators should be assigned to each group. While the participant groups rotate between all the stations, each facilitator guides one portion of the content area as outlined in the curriculum. Further research is needed to confirm the effectiveness of the use of station teaching with 12 participants in a group in the context of a low-resource setting. Furthermore, NEI enrolments should be informed by the contextual factors of the curriculum to guarantee that the number of students admitted into a programme corresponds to the strength and available resources of the institution. The restrictions must thus be carefully enforced by the MOH and its related agencies.

With reference to the use of SBCNE, this framework focuses on three outcome areas: participant, patient, and system. The benefits of simulation are well documented, including the development of cognitive, psychomotor, and affective skills of participants, as well as the enhancement of clinical competence, which leads to evidence-based practice and the effectiveness and efficiency of care provided to patients, resulting in positive patient outcomes and an effective health system. The majority of the favourable simulation outcomes documented are from first-world countries where simulation is largely used. Given that the implementation of SBCNE in low-resource settings is still a new notion, further research is needed to determine the benefits of simulation in the context.

Described as the mapping of curriculum content and learning objectives to simulation in the curriculum, curriculum integration is crucial for the successful implementation of simulation as an instructional method [8,30]. If SBCNE is well integrated into the curriculum, it promotes utilization of the concept [33]. This is particularly important for low-resource settings where the concept of SBCNE is new and may encounter some resistance from nurse educators and other members of the simulation community of learning. To ensure the effective design and implementation of SBCNE in low-resource settings, this framework therefore calls for a curriculum review to ensure the development of a competency-based curriculum with adequate integration of simulation. Similar to the transformation of nursing education and the inclusion of simulation in nursing curricula in other regions [33], we recommend the use of ‘experts’ with adequate knowledge in curriculum development and simulation to champion the curriculum review in NEIs in low-resource settings.

## 5. Conclusions

The user-centric approach adopted in the development of the framework and the inclusion of inputs from nurse educators, nursing students, and post-registration nurses contributed to the development of a framework that is comprehensive, context-specific, and tailored to the needs and resources of low-resource settings and capable of promoting access to and use of SBCNE. Even in high-income countries, the SBCNE framework can be used in low-resource settings. It is recommended that low-resource countries take seriously the adoption and implementation of this framework, as well as the use of SBCNE, in order to train competent nurses. This can be accomplished through close collaboration between the MOH, NMC, central government and their donor partners, and the leaders of NEIs by initiating policy directives and making available funds for the rapid scaling up of NEIs in low-resource settings. We therefore urge nursing leadership and nurse educator unions to take the lead in lobbying regulatory bodies, the central government, and their development partners to provide the necessary financial support and resources for the implementation of the framework and the adoption of SBCNE by NEIs in low-resource settings.

## 6. Strengths and Limitations

In advancing simulation theory, this study is the first to adopt a ‘user-centric’ sequential multimethod approach which blends personal experiences with findings of a scoping review as well as perspectives of nurse educators, students, and post-registration nurses in the framework development process. The choice of nominal group discussion (NGD) technique ahead of the Delphi technique in evaluating the framework was another novelty as it ensured the context specificity of the framework. The study is the first to provide a synthesis of constructs used to guide the design, implementation, and evaluation of simulation around the world, critically analysing their applicability in a low-resource setting using a scoping review.

Despite the measures adopted to ensure the rigour of the study, some limitations still remained. Although using the Delphi technique in addition to the NGD technique may have added more insight when participants evaluated the framework, it was not practicable due to a shortage of simulation experts in the research setting and the requirement to design a context-specific simulation framework. However, the informal solicitation of expert views on the draft framework minimized this limitation. It is impossible to say whether the results of the nominal group discussions evaluating the framework would have been different if all of the NCs in the research setting had been included in the process. Furthermore, due to resource limitations, the study was conducted in the diploma registered general nursing program of three diploma accredited public NCs and their primary clinical sites in Ghana. Therefore, there may be limitations in the application of the framework in other programs or other NEIs such as diploma in midwifery programs or baccalaureate programs. However, given the similarity of operations of the NEIs in low-resource stings and the contextual challenges they face, the framework may be applicable in other NEIs such as universities and in other low-resource settings.

## 7. Implications of the Study

The study is the first attempt at developing a context-specific simulation framework to promote the use of SBCNE in NEIs in low-resource settings. The SBCNE framework promises to be an effective tool to guide the introduction of SBCNE in low-resource settings. It is therefore imperative for low-resource countries to transform and scale up NEIs, which will guarantee the availability of infrastructure, essential logistics, and competent nurse educator workforce to support the use of the framework in facilitating the design, implementation, and evaluation of simulation. Further research could be conducted to implement and test the framework in the context of a low-resource setting and in other programs, such as diploma in midwifery or baccalaureate programs. Additionally, more research could be done to further explore the barriers to the implementation of SBCNE in the context of low-resource setting.

## Figures and Tables

**Figure 1 healthcare-10-01639-f001:**
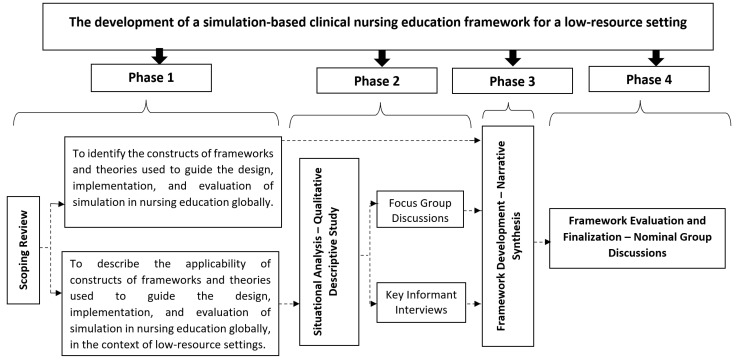
A flow chart of the sequential multimethod research design conducted in four phases.

**Figure 2 healthcare-10-01639-f002:**
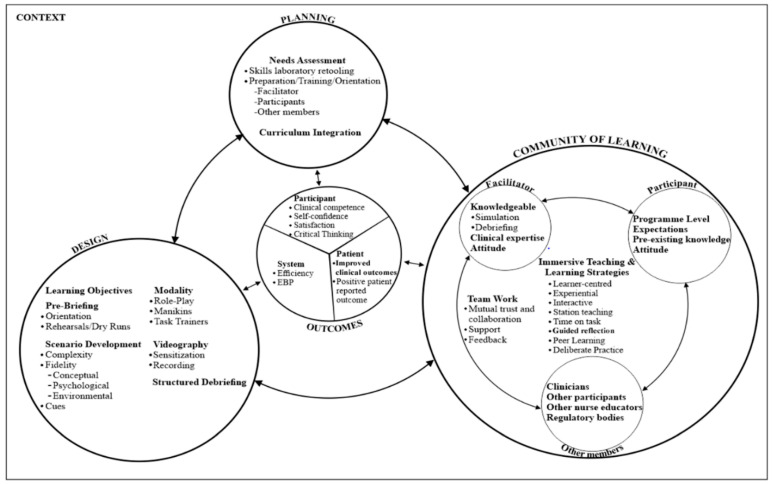
Simulation-based clinical nursing education framework for low-resource settings.

**Table 1 healthcare-10-01639-t001:** Situational analysis–qualitative descriptive study.

	Part 1–Focus Group Discussions	Part 2–Semi-Structured Interviews
**Population**	Post-registration diploma nurses within their first year of practice and nursing students (second- and third-year students) from six different sites (three NEIs which were all accredited diploma-awarding public nursing colleges and their primary clinical sites) in three geographical (northern, middle, and southern) zones of Ghana, a low-resource setting.	Nurse educators with full appointment and working in an accredited diploma-awarding public nursing college in the three geographical (northern, middle, and southern) zones of Ghana, a low-resource setting with three years’ experience in the teaching of practical skills and facilitating clinical competence development and fluent in English.
**Aim**	The study explored and described the experiences and perceptions of nursing students and post-registration nurses in the teaching and learning of clinical competence in Ghana, a low-resource setting.	To explore and describe the perceptions and challenges of nurse educators in in the teaching of practical skills and in facilitating the development of clinical competence in diploma nursing education in Ghana, a low-resource setting. Additionally, the study also explored and described the pedagogical strategies used by nurse educators in the teaching of practical skills and clinical competence development in Ghana, a low-resource setting.

**Table 2 healthcare-10-01639-t002:** Nominal group discussion guide.

**Question 1** Indicate below what you believe is/are the strength(s)/weakness(es) of this simulation-based clinical nursing education framework per the areas listed? When indicating the weaknesses of the framework, make suggestion in how to better the weaknesses.			
**Areas**	**Strength**	**Weakness**	**Suggestions**
**Structure** (*The graphical illustration of the phenomenon*)			
**Components** (*Inter-relations of the main and sub-components*)			
**Acceptability/Applicability**			
**Development process**			
**Question 2** (a) Which concept(s) do you believe should be removed or added to the simulation-based nursing education framework to make it applicable? (b) Provide the reason(s) for your answer **Question 3** Others: Provide further comment(s) you believe can help improve the applicability of the simulation-based nursing education framework.			

**Table 3 healthcare-10-01639-t003:** Demographic characteristics of participants.

Variables	Students	Post-Registration Nurses	Nurse Educators
**Gender**			
Male	5	4	1
Female	3	2	5
**Age (in years)**			
21–25	6	2	
26–30	2	4	
31–35	-	-	1
36–40	-	-	1
41–45	-	-	3
46–50	-	-	-
51–55	-	-	1
**Programme level**			
Second year	4	-	-
Third year	4	-	-
**Zone**			
Northern zone	3	2	2
Middle zone	3	2	2
Southern zone	2	2	2
**Work experience (in years)**			
1–5	-	6	2
6–10	-	-	3
11–15	-	-	-
16–20	-	-	1
**Highest academic qualification**			
Diploma	-	-	-
Bachelor’s degree	-	-	-
Master’s degree	-	-	5
MPhil	-	-	1
Doctorate degree	-	-	-
**Professional qualification in teaching**			
Diploma in Education	-	-	-
Bachelor of Education Health Sciences	-	-	3
Post-graduate diploma in education (PGDE)	-	-	1
Masters in Nursing Education			1
Master of Education			1

## Data Availability

Due to ethical approval conditions, the anonymized data pertaining to this study are available on request.

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
