# Peer review of "A Simulation-Based Clinical Nursing Education Framework for a Low-Resource Setting: A Multimethod Study"

_healthcare, 2022, doi:10.3390/healthcare10091639_

Round 1

Reviewer 1 Report (Previous Reviewer 2)

Good job. The authors have addressed my comments on the previous submission.  However, many of the findings are not supported by the data. See the result sections; content, planning, etc. 

Author Response

Reviewer 2 Report (New Reviewer)

This is a very interesting work that explores the possibilities of simulation-based learning in low-resource educational settings. Although it is a novel and interesting topic, methodologically it contains some aspects that can be improved:

1. The introduction deals extensively with the benefits of clinical simulation but is missing any allusion to its origins.

2. The objective of the study is not clearly defined.

3. The methodology, in particular phase 3-4, needs to be explained in more detail.

4. It would be necessary to describe the study population.

5. Include inclusion and exclusion criteria.

6. The results are profusely described, perhaps it would be necessary to restructure this section to make it shorter, including some table or graph that simplifies the information.

7. In the discussion it would be interesting to include the comparison with studies with similar characteristics, although the authors defend that there are no studies in low-resource settings.

8. A prospective research is missing. The authors state that more studies should be carried out but do not propose a future line of research.

Round 2

Reviewer 1 Report (Previous Reviewer 2)

Thank you for giving us this opportunity to review your revised paper! I have some comments as outlined below: 

1. A good rule of thumb for writing results sections is that each theme or sub-theme has at least two quotations from two different participants. Also, it’s better to not end with a quotation. Ending on quotations forces the reader to make the analysis and they may not have the same conclusion that you’d want them to have.

2.It is not clear from this description whether a subsequent meeting was convened between the research team and independent reviewers to discuss any possible discrepancies in data analysis and interpretation and, if so, how these were handled. Please elaborate here. 

Author Response

Please see he attachment.

Reviewer 2 Report (New Reviewer)

  All my comments on the previous text have been resolved. The suggested changes have improved the understanding of the
manuscript and I consider that it can be proposed for publication.

Author Response

This manuscript is a resubmission of an earlier submission. The following is a list of the peer review reports and author responses from that submission.

Round 1

Reviewer 1 Report

Thank you for the opportunity to read your work and provide review. And I apologize for my delay with the return of this review. This manuscript was well written and addresses the important topic of preparing nursing students for practice. I have included some opportunities for revision and clarification in order to more prominently highlight the work and increasing reader understanding.

Abstract:

·       Lines 17-25 are a repeat of the ones above.

·       Suggest condensing the explanation of the model starting on line 25, to a single sentence and then expand in manuscript.

·       Lines 33-36 provide an actionable implication of the work.

Introduction

·       The introduction presents a well-reasoned argument for the need (gap) of this work, particularly lines 74-78.

Methods

·       This manuscript describes a complicated project with a sequential multimethod design. A graphic of the design could be a concise way to present the information on lines 83-106.  Since you describe the steps in the next sections, a simple chart or graphic would be sufficient.

·       Suggest restating/stressing that the methods were iterative

·       Suggest rephrasing lines 103-106 for clarity- it is an important sentence and would be more impactful if clearer.

·       Section 2.2 Phase 1: Scoping Review

o   Describe how the scoping review was done- what was the search strategy?

o   And then move the results (lines 122-123) to results section.

·       Section 2.3

o   Lines 126-130 could also be presented in a simple chart/graphic

o   Aim starting on line 131 is well-stated,

o   Suggest using the phrase “qualitative descriptive design” earlier than line 134/135 or replacing it with “situational analysis” to be consistent.

o   Move results starting at line 135 to Results section

Results

·       Suggest beginning the results section with the findings previously mentioned in methods section (above)

·       Clearly state that in section 3.1 you are introducing the first draft of the model, which is then evaluated with the NGD and those results were organized into three categories. (If I am interpreting that process correctly)

·       Each subsections of 3.2 could be condensed into one section concise section. For example, Section 3.2.1 could be condensed into one section that addresses Appropriateness, Acceptability, and Applicability.

·       Clarify that section 3.3 is the final, proposed framework informed by the work described previously. The importance of this final framework was diluted, showcase it! Outright state- here is our final product.

·       Suggest condensing sub-subsections again; some of the content from these sections could be moved to the discussion.  Tightening up the descriptions in the results sections would make the section more impactful, again highlight the good work, try not to dilute it with extras.

·       Lines 426-427 not necessary

·       The paragraph starting at line 454- did this recommendation come out of the NGD? Or a suggestion of authors, if so- move to discussion.

·       Sentence at line 500- is this true for all clinical practice vs simulation? Clarify this comment and/or move to discussion.

·       Line 524 instead of “friendly” perhaps use trusting, collegial, mutual support (which is mentioned at line 612)

·       Line 528- suggest rephrasing “To guarantee a meaningful simulation”

Discussion

·       Paragraph starting at line 586 provides a nice rationale for the work and need for the model

·       Either in the discussion or conclusion sections, follow back up with the concept of competency-based curricula since it was introduced in the introduction.

·       Was there any discussion about using standardized patients? Depending on the purpose, it could be costly, but could also explore ways to engage community (patients, caregivers, advocates) or even other student groups to play roles. *if appropriate in Ghana, I am not very familiar. For example, we have had a real patient who uses a wheelchair play our “patient” during a simulation. It added a level of reality for the nursing students.

Implications

·       Include what are the next for the model- implementation, testing?

Author Response

Kindly find attached, response to reviewer comments for your attention. 

Reviewer 2 Report

Thank you for the opportunity to review this interesting paper, which describes a study of A Simulation-Based Clinical Nursing Education Framework for 2 Low-Resource Settings
I have divided up my comments based on the major headings of the paper for clarity.

Abstract: abstract is clear and concise.

Introduction: 

The literature review demonstrates what is known and not known and identifies the need for the study to be conducted - good.

The methods are generally well described.  There are a few details that will make this section more complete:
-Design: This is really important:  Can you add a sentence or two that describes why a sequential multimethod research design
 was used in this study? 

What was the relationship between the researchers and participants? This is important in qualitative research because it often affects who chooses to participate in the study and who does not, which affects the results potentially.

Where were the interviews conducted (setting)? Individually? In a private, quiet place?
any pilot testing of the interview questions? and where is the interview guide in the paper?.
-Were any repeat interviews done to clarify what the participants originally said?
-Were any field notes taken?
-Need to add additional detail at the end of the methods section about the trustworthiness of the study (credibility, dependability, transferability, and confirmability).
-Was any member checking done once themes were derived?

Thank you again for the opportunity to review your work, and I hope my comments and suggestions are helpful as you work on this.

Reviewer 3 Report

There are parts of the abstract that overlap(line 10-17/line 17-25).

This study is an attempt to find ways to provide quality education to students. Because the approaches of quantitative and qualitative research are different, you can help readers understand the paper by clearly describing the main concepts and categories and explaining the theoretical framework while describing the research results.

Please describe the analysis method in detail.

Please refer to the author guidelines and make corrections to the contents of the paper.

Round 2

Reviewer 3 Report

Because the content of the dissertation is vast, it is difficult to convey the core content well. Please write concisely so that readers can understand the conclusion of your paper. Overall, papers that do not follow the author guidelines are difficult to publish.

Please upload the word file in the 'Apply all changes and stop tracking' status. It is also necessary to review the research methodology. Does focus group discussion mean focus group interview (FGI)? Proper use of terminology is required to become scientific knowledge.